Identification and characterization of the karrikins signaling gene SsSMAX1 in Sapium sebiferum

Ni Fang 1
Shah Faheem Afzal 2
Ren Jie 2 rejieaaas@sina.com
1 Anhui Wenda University of Information Engineering , Hefei, Anhui , China
2 Institute of Agricultural Engineering, Anhui Academy of Agricultural Sciences , Hefei, Anhui , China
Abd El-Moneim Diaa
Electronic publication date: 2023 Dec 8
Publication date: 2023
Volume: 11
Electronic Location ID: e16610
Received 2023 May 31; Accepted 2023 Nov 15
Copyright: © 2023 Ni et al.
Copyright year: 2023
Copyright holder: Ni et al.
License: This is an open access article distributed under the terms of the Creative Commons Attribution License, which permits unrestricted use, distribution, reproduction and adaptation in any medium and for any purpose provided that it is properly attributed. For attribution, the original author(s), title, publication source (PeerJ) and either DOI or URL of the article must be cited.
License URL: https://creativecommons.org/licenses/by/4.0/

Keywords: SMAX1, Seed dormancy, Seedlings development, Hypocotyl length, Root length

Funding: Landscape Application and Rapid propagation of Strong Seedings of Hydrangea in Anhui province KJ2020A0815, XZR2020A03 This work was supported by Research on the Landscape Application and Rapid propagation of Strong Seedings of Hydrangea in Anhui province (Project No: KJ2020A0815; XZR2020A03). The funders had no role in study design, data collection and analysis, decision to publish, or preparation of the manuscript.

==============================
SUPPRESSOR OF MAX2 LIKE 1 (SMAX1) is a member of the SUPPRESSOR of MAX2 1‑LIKE family of genes and is known as a target protein of KARRIKIN INSENSITIVE2 (KAI2)-MORE AXILLARY BRANCHES2 (MAX2), which mediates karrikin signaling in Arabidopsis. SMAX1 plays a significant role in seed germination, hypocotyl elongation, and root hair development in Arabidopsis. SMAX1 has not yet been identified and characterized in woody plants. This study identified and characterized SsSMAX1 in Sapium sebiferum and found that SsSMAX1 was highly expressed in the seed, hypocotyl, and root tips of S. sebiferum. SsSMAX1 was functionally characterized by ectopic expression in Arabidopsis. SsSMAX1 overexpression lines of Arabidopsis showed significantly delayed seed germination and produced seedlings with longer hypocotyl and roots than wild-type and Atsmax1 functional mutants. SsSMAX1 overexpression lines of Arabidopsis also had broader and longer leaves and petioles than wild-type and Atsmax1, suggesting that SsSMAX1 is functionally conserved. This study characterizes the SMAX1 gene in a woody and commercially valuable bioenergy plant, Sapium sebiferum. The results of this study are beneficial to future research on the molecular biology of woody plants.

Introduction

Karrikins (KARs) are butenolides, which are formed by the burning of biomass. Karrikins enhance seed germination (Nelson et al., 2009; Flematti et al., 2004; Nelson et al., 2012), promote seedling response to light in the modal Arabidopsis thaliana, and improve seedling vigour in many plants (Jain, Kulkarni & van Staden, 2006; Kulkarni et al., 2006; Nelson et al., 2010; van Staden et al., 2006). In plants, the putative receptors for KARs and Strigolactones (SLs) are the paralogues a/b-hydrolases KARRIKIN INSENSITIVE2 (KAI2) and DWARF14 (D14), respectively. These paralogues are ancient and exist in all angiosperms (Waters et al., 2012, 2013). KAI2 and D14 require common catalytic triad residues (Ser95-His246-Asp217) for binding to KARs and SLs, respectively (Hamiaux et al., 2012; Hidemitsu et al., 2013; Li et al., 2013; Megumi et al., 2013; Rohan et al., 2013; Toh et al., 2015; Xia et al., 2013). SL and KAR signaling are both dependent on the activity of MAX2, an F-box containing protein that forms an Skp-Cullin-F-box (SCF) complex (Nelson et al., 2011; Stirnberg, Furner & Ottoline Leyser, 2007; Stirnberg, van De Sande & Leyser, 2002). The SCF complexes have ubiquitin moieties that bind to the target proteins, degrading the target proteins by the 26S proteasome (Somers & Fujiwara, 2009). Ligand binding or hydrolysis in MAX2 might encourage conformational receptor variations, which may change MAX2’s interactions with downstream signaling protein partners (Hamiaux et al., 2012). The functional mutation of Atmax2 showed various phenotypes, such as delayed seed germination, elongated hypocotyl, narrow leaves, and more axillary branches in Arabidopsis. The phenotype differences between strigolactones insensitive D14 (more axillary branches) and KARRIKINS INSENSITIVE 2 (KAI2) phenotypes (seed germination, hypocotyl length, and leaf width) indicate that SLs and KARs control separate aspects of MAX2-dependent functions (Waters et al., 2012). These studies all suggest that the karrikin and strigolactone signaling pathways are MAX2-dependent but regulate different aspects of plant life.

The karrikins signaling pathway requires the deactivation of repressor proteins following hormone perception by ubiquitylation and subsequent proteasomal degradation (Nelson et al., 2012, 2010, 2011; Soundappan et al., 2015; Stanga, Morffy & Nelson, 2016; Stanga et al., 2013; Waters et al., 2012). Forward genetic screening has identified KARs signaling repressors (Soundappan et al., 2015; Stanga, Morffy & Nelson, 2016; Stanga et al., 2013; Wang et al., 2015; Zhou et al., 2013), which are members of the same SUPPRESSOR OF MAX2-like (SMXL) family. The SMXL family includes three clades in seed plants: (1) SMAX1 and SMXL2 for KARs signaling, (2) SMAX 6, SMAX 7, and SMAX 8 for strigolactones signaling, and (3) SMAX3, SMAX4, and SMAX5 acting in an unidentified signaling module that regulates phloem formation but is independent of strigolactones and KARs signaling (Soundappan et al., 2015; Stanga, Morffy & Nelson, 2016; Stanga et al., 2013; Walker et al., 2019; Wallner et al., 2017; Wang et al., 2015). The strigolactones signaling repressors, such as SMXL6, SMXL7, and SMXL8, and D53 proteins are ubiquitylated and degraded by proteasomes upon ligand sensing (Jiang et al., 2013; Soundappan et al., 2015; Walker et al., 2019; Zhou et al., 2013). It has also recently been discovered that SMAX1 and SMXL2 proteins are degraded by the interaction of KAI2 and MAX2 proteins (Khosla et al., 2020; Wang et al., 2020). The role of SMAX1 and SMXL2 in plant development has been genetically studied in Arabidopsis, where smax1smxl2 double mutants display mild, kai2-opposing phenotypes, such as slightly faster seed germination, shorter hypocotyls, bigger cotyledons, and longer root hairs (Stanga, Morffy & Nelson, 2016; Villaécija-Aguilar et al., 2019). The SUPPRESSER OF MAX2 LIKE 1 (SMXL1) protein is the third class of putative target genes that have been identified. Functional mutant screening for SUPPRESSOR OF MAX2-like proteins in Arabidopsis led to the discovery of SMAX1, which regulates seed germination, seedling growth, leaf shape, and size (Stanga et al., 2013). A recent study provided a detailed description of SMXL family genes in a model woody plant, P. trichocarpa, but only characterized PtSMXL7, a signaling gene of strigolactones (Sun et al., 2023). The characterization of SMAX1 in woody plants has not yet been investigated.

Chinese tallow (Sapium sebiferum L.), which belongs to the Euphorbiaceae family, is native to eastern Asia (Esser, 2002). Its fruits produce a highly-saturated fatty acid in the tallow layer and highly-unsaturated oil in the seed (Boldor et al., 2010). Tallow is used for manufacturing candles, soap, cloth, and fuel, and the seed oil can be used for making varnishes and paints (Brooks et al., 1987; Jeffrey & Padley, 1991). A single mature S. sebiferum tree produces many seeds, and is estimated to produce 4,700 liters of oil per hectare, annually. The average commercial yields of S. sebiferum far exceed traditional oilseed crops (Boldor et al., 2010; Webster, Jenkins & Jose, 2006). S. sebiferum is also popular because of its colorful autumn foliage, and has become a popular species for landscaping and as a source of biodiesel (Gao et al., 2016).

S. sebiferum seeds are dormant and require long stratification times to start normal germination. Seed germination, which is essential to the plant life cycle because it determines plant survival and reproductive success, is regulated by different hormones and signaling compounds, such as ABA, GA, ethylene, and karrikins (Carbonnel et al., 2020; Dekkers et al., 2016; Lee et al., 2002; Nelson et al., 2012). Seed culturing is an easy and widely-used method of commercial propagation of many plant species, including many bio-energy plants, so an efficient seed germination assay is needed by both researchers and nursery growers. S. sebiferum is best propagated through the seed, but S. sebiferum’s poor seed germination rate due to deep dormancy has limited the use of this species (Conway, Smith & Bergan, 2000). Only a few studies are available to understand and promote S. sebiferum seed germination. One study found that S. sebiferum seeds exposed to cold water for 72 h showed 10% seed germination (Li et al., 2012). Another study found that the dormancy of tallow tree seeds could be overcome by soaking the seeds in GA3, followed by 100 days of cold stratification (Li et al., 2012). A separate study explored the effect of KAR1 on Sapium sebiferum seed germination (Shah et al., 2020). In the karrikins pathway, SMAX1 has been found to be abundant in seeds and to play a significant role in seed germination and seedling development. Despite this discovery, it is still unknown whether SMAX1 functions are conserved in a perennial woody plant like S. sebiferum. This study analyzed the ectopic expression of SsSMAX1 in Arabidopsis to confirm whether the functions of SsSMAX1 are conserved. SSMAX1 was identified by blasting AtSMAX1 into a local blast library created by an already available S. sebiferum transcriptome. Ectopic expression of SSMAX1 in Arabidopsis was phenotypically compared with loss-of-function mutants of SMAX1 and wild-type Arabidopsis.

Materials and Methods

Bioinformatic analysis

The 3D structure of the SsMAX1 protein was predicted by the Phyre2 web portal for protein modeling, prediction, and analysis (http://www.sbg.bio.ic.ac.uk/phyre2; Kelley et al., 2015). The Arabidopsis thaliana SMAX1 (AT5G57710) gene sequence was downloaded from TAIR (https://www.arabidopsis.org). Arabidopsis thaliana, Populus trichocarpa (XP_002324496.2), and Sapium sebiferum SMAX1 gene sequences were aligned using the residue substitution matrix in the AlignX in Vector NTI Advance 11.0. For phylogenetic tree construction, Arabidopsis (AT5G57710), P. trichocarpa (XP_002324496.2), Nicotiana tobacum SMAX1 (XP_016511300.1), Carica papaya SMAX1 (XP_021895142.1), Oryza sativa D53 (NP_001410055.1), and Sapium sebiferum SMAX1 protein sequences were aligned using the default settings of the ClustalW in MEGA11 software. After the alignment of proteins, the alignment results were saved in MEGA format, and a phylogenetic tree was developed by a neighborhood joining method using the Poisson model with 500 bootstrap replications in MEGA11 software.

Cloning of Sapium sebiferum SMAX1 gene

A local blast library was built in NCBI blast-2.2.31 using a fasta file of S. sebiferum flower-bud transcriptome (accession: SRX656554; Yang et al., 2015). After constructing the local blast library, the complete sequence of the SsSMAX1 gene was obtained using the Arabidopsis SMAX1 amino-acids sequence as an input query in the tblastn function of NCBI blast-2.2.31 (Camacho & Madden, 2013). The Arabidopsis SMAX1 (AT5G57710) amino-acids sequence was obtained from TAIR (https://www.arabidopsis.org/). The full-length sequence of SsSMAX1 with the translated amino acid sequence is given in Table S1. The full-length open reading frame (ORF) of the SsSMAX1 gene was found using the NCBI ORF finding tool. Gene-specific primers were designed by Primer Premier 5 to amplify the full-length ORF of SsSMAX1. The Tm of the primers was between 62.0 and 65.0 ° C; a list of all primers is given in Table S1. The gene cloning method is fully described in Fig. S1.

Expression vector design

After cloning the full-length sequence of SsSMAX1, the PCR product was aligned with a pEASY®-Blunt Cloning Vector (TransGen Biotech Co., Beijing, China). The ORF of the SsSMAX1 gene was sequenced and confirmed by Sangon Biotech (Shanghai) Co., Ltd. The cloned gene sequence in a pEASY®-Blunt Cloning Vector was double digested at Sal1 from the start, and Sma1 from the stop codon site, and then inserted into the expression vector, pOCA30, which was also double-digested at the same restriction sites. A map of the expression vector is drawn in Fig. S1. The expression vector was transformed to the Agrobacteria EHA105 strain. The floral dip method was used for gene transformation in Arabidopsis (Zhang et al., 2006). Seeds of the T4 homozygous SsSMAX1 line (two lines) were selected for further experiments.

Plant materials and growth conditions

Arabidopsis wild-type Columbia-0 (Col-0) and mutant Atsmax1 were obtained from the Arabidopsis Biological Resources Center (Columbus, Ohio). Seeds of the T4 homozygous SsSMAX1 lines (two lines) were selected for further experiments. Seeds of wild-type plants, Atsmax1 and SsSMAX1, were surface sterilized by serial washing with 70% (v/v) ethanol for 2 min, then with 10% (v/v) NaClO for 10 min, and then washed three times with double distilled water. The sterilized seeds were plated on ½ Murashige and Skoog (MS) medium supplemented with 1% (w/v) sucrose plus 0.8% (w/v) agar and placed at 4 °C for 2 days. Seeds were germinated in a 16 h/8 h photoperiod at 22 °C, approximately 150 mmol photons m2/s2. Seven-day-old Arabidopsis seedlings were transferred from ½ MS medium to soil and grown in a growth room at 22 °C, approximately 150 mmol photons m2/s2 with 16 h/8 h (long-day conditions) photoperiods.

Seed germination, hypocotyl length, and growth parameters analysis

The seed germination analysis was performed in 10 × 10 Petri plates. Each experiment was repeated five times, with 30 seeds of each genotype. Germinated seeds were counted starting 24 h after seeds were placed in a growth room. For hypocotyl and root length measurement, seeds were sown in 10 × 10 plates with grids. Pictures were taken after seven days of seed germination; ImageJ software measured root and hypocotyl length (15 seedlings). Leaf length (15 leaves per plant) was measured using ImagJ by taking leaf pictures of three different 45-day-old plants.

Primer design, RNA extraction, cDNA synthesis, and RT–qPCR conditions

SsSMAX1 gene’s full-length CDS and protein sequences are available in Document S1. Primers used for qPCR were designed with Primer Premier 6. The Tm of the primers was between 59.0 °C and 61.0 °C; a list of all primers is given in Table S1. For the gene expression analysis in the different tissues of S. sebiferum, samples (three biological replicates) were taken from the S. sebiferum tree growing at Anhui Academy of Agricultural Sciences. To measure SsSMAX1 expression in overexpression lines, leaf samples (three biological replicates) were taken from 15-day-old seedlings of WT, OE1, and OE2 Arabidopsis. Samples were frozen in liquid nitrogen and stored at −80 °C. RNA was extracted using an E.Z.N.A® plant RNA extraction kit (OMEGA Pro -TEK) based on the manufacturer’s instructions. Five hundred ng RNA of each sample was reverse transcribed using cDNA synthesis SuperMix (TransGen Biotech, Beijing, China) based on the manufacturer’s instructions. The cDNA samples were diluted 25X with sterile water. For each 20 microliter (μL) reaction of qPCR, 9 μL of cDNA, 10 μL of the 2X QuantiNova SYBR Green PCR Master Mix (QIAGEN, Hilden, Germany), and 0.5 μL of each primer were added to a final total volume of 20 μL. The qRT–PCRs were run on a Light Cycler®96 (Roche, Basel, Switzerland). The qPCR program consisted of two steps: the first step at 95.0 ° C for 3 min, and the second step had 45 cycles alternating between 15 s at 95.0 °C, 15 s at 60.0 °C, and 15 s at 72.0 °C.

Statistical analysis

Excel 2013 was used to arrange the data and R Studio 1.1.383 was used for the statistical analyses. The data were represented as mean ± standard deviation. Results from the different treatments were analyzed separately. The wild-type Arabidopsis was used as the control group for each set of experiments. The results of overexpression lines (OE1 and OE2) and Atsmax1 were compared with the results of wild-type Arabidopsis. The significance of treatments was tested by one-way analysis of variance (ANOVA). Tukey tests were used to identify significant differences between pairs of means at p < 0.05.

Results

Bio-informatics analysis of SsSMAX1

SsSMAX1 showed significant alignment with the SMAX1 proteins of modal plants like Arabidopsis thaliana and Populus trichocarpa (Fig. 1A). The three-dimensional structure showed that SsSMAX1 contained the loops and turns, alpha helix, and beta sheets (Fig. 1B). Protein alignments calculated by residue substitution matrix showed that SsSMAX1 is 65.1% similar to AtSMAX1 and 80.4% identical to PtSMAX1 (Fig. 1C). The phylogenetic tree showed that SsSMAX1 has the same clad as papaya CpSMAX1; these proteins lay subclade to Nicotiana tobacum SMAX1. SsSMAX1 was a neighbour to AtSMAX1 (Fig. 1D).

Figure 1 Bio-informatics analysis of SsSMAX1.

(A) Sapium sebiferum SMAX1 protein alignment with Arabidopsis and Populus SMAX1 protein. (B) Similarity and identity of SsSMAX1 with AtSMAX1 and PtSMAX1. (C) The ribbon diagram of the tertiary (three-dimensional) structure of SsSMAX1. (D) Phylogenetic tree of SsSMAX1 protein.

Figure 2 SsSMAX1 relative expression in different organs and tissue in Sapium sebiferum.

Gene expression determination by qPCR. Data represent the 2−∆∆Ct value of each gene. SsACTIN2 was used as a reference gene. Data were statistically analysed by one-way ANOVA, and multiple comparisons were made with HSD Tuckey’s test at p = 0.5 significant level (n = 3).

SsSMAX1 distribution in Sapium sebiferum

The molecular function of a gene can be predicted by its expression levels in different tissues of Sapium sebiferum. The results of an analysis of expression levels showed that SsSMAX1 was highly expressed in hypocotyl, root tip, shoot tip, and young leaf in Sapium sebiferum (Fig. 2), indicating that SsSMAX1 is involved in the regulation of leaf shape, hypocotyl development, and root architecture development.

SsSMAX1 involvement in seed germination and seedling development

SsSMAX1 was found to be overexpressed in the model plant Arabidopsis thaliana. Results showed that the ectopic expression of SsSMAX1 in Arabidopsis hindered the seed germination frequency of the transgenic SsSMAX1 phenotype (Fig. 3A). In contrast, Atsmax1 function mutants showed quicker germination than wild-type and SsSMAX1 background seeds (Fig. 3B). Hypocotyl length was increased in SsSMAX1 background Arabidopsis seedlings compared to wild-type and Atsmax1 seedlings, and Atsmax1 had the shortest hypocotyl length of the three (Figs. 3C and 3D).

Figure 3 SsSMAX1 involves seed germination, hypocotyl and root development.

(A) Seed germination of SsSMAX1 transgenic lines; overexpression line 1 (OE1) and 2 (OE2), Atsmax1, and wild-type (WT) Arabidopsis. Black arrows head represent non-germinated seeds. (B) Graphical presentation of seed germination result of part A, multiple comparisons were made with HSD Tuckey’s test at p = 0.5 significant level (n = 5). (C) Hypocotyl of wild-type, Atsmax1, OE1 and OE2 Arabidopsis lines. White arrows are indicated root length difference of Atsmax1 is greater than the roots of other genotypes. (D) Graphical ANOVA of (C). (E) Root length of WT, Atsmax1, OE1 and OE2 after 7 days of germination. All data were analysed with one-way ANOVA, and multiple comparisons were made with HSD Tuckey’s test at p = 0.5 significant level (n = 15). (A) white bar = 2 cm. (C) white bar = 1 cm.

SsSMAX1 expression in Arabidopsis and its role in plant growth and development

The expression level of SsSMAX1 was evaluated in Arabidopsis by qPCR, and the results showed that SsSMAX1 was highly expressed in both transgenic lines. Expression levels of AtSMAX1 were considerably lower than SsSMAX1 (Fig. 4). A phenotypical analysis showed that SsSMAX1 and wild-type had more rosette branches than Atsmax1. Total number of cauline leaves, inflorescence height, and number of secondary branches were not significantly different between SsSMAX1, Atsmax1, and wild-type. Leaf width was remarkably higher in Atsmax1 than in SsSMAX1 and wild-type (Figs. 5A–5D). Leaf length and petiole length were significantly larger in SsSMAX1 (Figs. 5A–5D).

Figure 4 Expression level of AtSMAX1 and SsSMAX1 in Arabidopsis.

The figure shows the expression levels of AtSMAX1 and SsSMAX1 in wild-type (WT) and SsSMAX1 transgenic lines (OE1 and OE2, respectively) of Arabidopsis. Leaves of 15-day-old seedlings of wild-type (WT) and SsSMAX1 transgenic Arabidopsis line 1 (OE1) and 2 (OE2) were collected from gene expression analysis. All data were analysed with one-way ANOVA, and multiple comparisons were made with HSD Tuckey’s test at p = 0.5 significant level (n = 3).

Figure 5 SsSMAX1 involved in plant leaf development.

(A) Leaves of 35-day-old wild-type seedlings (WT), Atsmax1, and SsSMAX1 transgenic Arabidopsis lines (OE1 and OE2). The white bar on the basement of the picture = 1 cm. (B–D) Petiole length, leaf length and leaf width, respectively, cm is centimetre. All data were analysed with one-way ANOVA, and multiple comparisons were made with HSD Tuckey’s test at p = 0.5 significant level (n = 15). (A) white bar =1 cm.

Discussion

S. sebiferum has mainly been propagated from seed, but because of deep dormancy, the effectiveness of this strategy has been severely constrained. Previous studies found that KAR1 impacts Sapium sebiferum seed germination, and that SMAX1 is widely distributed in seeds and is essential for seed germination and seedling growth in the karrikins pathway. Recent studies suggest that the homologous SUPPRESSOR OF MAX2 1 (SMAX1) in Arabidopsis and DWARF53 (D53) in rice (Oryza sativa) are downstream targets of MAX2 (Jiang et al., 2013; Soundappan et al., 2015; Zhou et al., 2013). SMAX1 is one of the eight SMXL family proteins involved in KARs and SLs signaling (Soundappan et al., 2015; Stanga et al., 2013). The results of a previous study showed that SMAX1 is a downstream target of KAI2-MAX2-mediated signaling of karrikins (Soundappan et al., 2015), but it is still unclear whether SMAX1 functions are preserved in a perennial woody plant like S. sebiferum. This study identified SMAX1 in S. sebiferum and then functionally characterized it by ectopic expression of SsMAX1 in Arabidopsis thaliana.

A bioinformatics analysis revealed that SsSMAX1 is more similar to PtSMAX1 than to AtSMAX1, and the phylogenetic tree showed that SsSMAX1 has the same clad as papaya CpSMAX1. Both proteins lie subclade to Nicotiana tabacum SMAX1, which predicted the conserved functions of SsSMAX1. Because SsSMAX1 is more similar to PtSMAX1, they may have similar roles. Previous studies have shown that Populus trichocarpa SMAX1 (PtSMAX1) and other proteins are functionally conserved (Sun et al., 2023; Tang et al., 2010). In this study, SsSMAX1 expression was checked in different parts of the S. sebiferum plant, and SsSMAX1 was phenotypically characterized in Arabidopsis. The expression pattern of SsSMAX1 revealed that it is abundantly expressed in seeds, hypocotyl, and root tips, which is consistent with the findings of previous studies (Nelson et al., 2010; Soundappan et al., 2015; Stanga et al., 2013; Villaécija-Aguilar et al., 2019). These results suggest that SMAX1 is involved in seed germination, hypocotyl, and root development.

Karrikins was discovered to be involved in Arabidopsis seed germination, which requires gibberellic acid synthesis and light. In this study, SsSMAX1 was ectopically expressed in Arabidopsis with the help of a 35S promoter, and seeds of SsSMAX1 background Arabidopsis were slightly more dormant than wild-type Arabidopsis seeds. The results of a comparison between the seed germination of SsSMAX1 phenotypes and the Atsmax1 functional mutants revealed that Atsmax1 was non-dormant compared to SsSMAX1 seeds. These results are consistent with Stanga et al. (2013), who reported that the Atsmax1 mutant’s seed could germinate early and suppress the dormancy created by the functional mutation of Atmax2. A previous study showed that ABA removes the acceleration effect of KARs on germination, and KARs need the biosynthesis of GA to promote seed germination (Nelson et al., 2009). KARs suppress the expression of IAA1, which is the IAA response gene; thus, KARs may accelerate seed germination by suppressing the signals of IAA (Nelson et al., 2011). The results of this study showed that ectopic expression of SsSMAX1 in Arabidopsis promoted dormancy in seeds, suggesting that the function of SsMAX1 is conserved in seed germination.

Karrikin signaling gene SMAX1 has been reported to regulate Arabidopsis hypocotyl elongation by regulating auxin homeostasis (Soundappan et al., 2015). The results of this study showed that the seedlings of SsSMAX1 OEs have longer hypocotyl than wild-type Arabidopsis. These results support the findings of previous studies that SMAX1 regulates hypocotyl length and is downstream of KAI2-MAX2-mediated signaling of karrikins (Soundappan et al., 2015; Xu, Jinbo & Cai, 2022). These results suggest that the hypocotyl elongation function of SsSMAX1 is similar to the SMAX1 gene of other candidate species, such as Arabidopsis.

Leaf development, such as leaf width and petiole length, has also been reported as a function of SMAX1. The results of this study showed that SsSMAX1 background Arabidopsis has more extended and less broadleaf than wild-type and Atsmax1 functional mutants. These results are consistent with the results of previous studies, which demonstrated that karrikins Atsmax1 produced seedlings that had broader leaves than wild-type (Nelson et al., 2012; Soundappan et al., 2015; Stanga et al., 2013). The petiole length of SsSMAX1 plants was much longer than wild-type and AtSMAX1 functional mutants. These results are consistent with the results of Stanga et al. (2013) that the enlarged length of the petiole of Atmax2 was suppressed in Atsmax1-Atsmax2 double mutant seedlings. A previous study found that the accumulation of SMAX1 promotes the leaf length/width ratio and petiole length by regulating the expression of the genes involved in auxin transport, the cytokinin signaling pathway, and SL biosynthesis (Zheng et al., 2021). Root growth is necessary to expand the absorptive root surface area and plays a crucial role in plant life. The function of SsMAX1 has been reported in the development of roots and root hairs. This study showed that ectopic expression of SsSMAX1 reduced root length and root hairs in Arabidopsis, in line with previous studies that showed SMAX1 regulates root hair elongation by repressing ethylene biosynthesis via inhibition of PIN2 (Carbonnel et al., 2020; Villaécija-Aguilar et al., 2019, 2022). This study’s results suggest that Sapium sebiferum SMAX1 regulates seedling development, including seed germination, hypocotyl development, root length control, and leaf length and width regulation, thus, SsSMAX1 is also functionally conserved.

Conclusion

This is the first study to characterize the SMAX1 of woody perennial plant, S. sebiferum. In alignment with previous studies of different plant species (Carbonnel et al., 2020; Jiang et al., 2013; Soundappan et al., 2015), SsSMAX1 regulated seed germination, hypocotyl length, petiole length, leaf length and width, and root length in Arabidopsis, revealing that SMAX1 has conserved functions in S. sebiferum. This study also provides the foundation for an understanding of the role of SMAX1 in other plant species.

Supplemental Information

Supplemental Information 1 SsSMAX1 gene’s full-length CDS and protein sequences.

SsSMAX1 gene’s full-length CDS and protein sequences

Click here for additional data file.

Supplemental Information 2 List of Primers.

Primers used for qPCR were designed by using primer Premier 6. The Tm of the primers was between 59.0 and 61.0° C . Primers used for qPCR were designed by using primer Premier 5.

Click here for additional data file.

Supplemental Information 3 Supplementary Figure 1: Diagram of over-expression vector.

SsSMAX1 was cloned by using primers given in supplementary table 1. Ligated to blunt end vector, sequenced, and then digested at Sal1 and Sma1 positions. Overexpression vector pOCA30 was also double-digested at Sal1 and Sma1 positions. The Full-length SsSMAX1 gene was ligated to pOCA30.

Click here for additional data file.

Supplemental Information 4 Figure 2 data.

Gene expression determination by qPCR. Data represent the 2−∆∆Ct value of each gene. SsACTIN2 was used as a reference gene.

Click here for additional data file.

Supplemental Information 5 Figure 3 data.

Hypocotyl length, Root length, Seed germination of SsSMAX1 transgenic lines; overexpression 1 (OE1), 2 (OE2), Atsmxl1, and wild-type (WT)

Click here for additional data file.

Supplemental Information 6 Figure 4 data.

Expression levels in wild-type and SsSMAX1 transgenic line 1 and line 2 (OE1 and OE2 respectively). Leaves of 15-day-old seedlings of wild-type (WT), AtSMAX1, and SsSMAX1 transgenic Arabidopsis line 1(OE1) and 2 (OE2) were collected from gene expression analysis.

Click here for additional data file.

Supplemental Information 7 Figure 5 data.

Leaf length, Leaf width, and Petiol length of SsSMAX1 transgenic lines; overexpression 1 (OE1), 2 (OE2), Atsmxl1, and wild-type (WT)

Click here for additional data file.

Additional Information and Declarations

Competing Interests

Author Contributions

DNA Deposition

Data Availability

The authors declare that they have no competing interests.

Fang Ni conceived and designed the experiments, performed the experiments, analyzed the data, prepared figures and/or tables, and approved the final draft.

Faheem Afzal Shah performed the experiments, analyzed the data, prepared figures and/or tables, authored or reviewed drafts of the article, and approved the final draft.

Jie Ren conceived and designed the experiments, analyzed the data, authored or reviewed drafts of the article, and approved the final draft.

The following information was supplied regarding the deposition of DNA sequences:

The sequences are available at GenBank: SRX656554.

The following information was supplied regarding data availability:

The raw measurements are available in the Supplemental Files.

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
