# Peer review of "Identification and characterization of the karrikins signaling gene SsSMAX1 in Sapium sebiferum"

_PeerJ, doi:10.7717/peerj.16610_

## Round 0.1 · original submission · Major Revisions

Dear Authors
According to the reviewer's comments, this manuscript cannot be accepted for publication in its current form. It needs a major revision to be reconsidered for publication. The authors are invited to revise the paper, considering all the suggestions made by the reviewers. Please note that requested changes are required for publication.
With Thanks

·

Basic reporting

Based on the provided SMAX text, the article demonstrates a sufficient level of clarity and unambiguity in presenting the information. The sentences are well-structured, and the overall language used is appropriate for scientific writing. The introduction provides a brief overview of Karrikins (KARs) and their role in seed germination and seedling development, mentioning the putative receptors for KARs and SLs. The background information could be expanded to include more details about the significance of SMAX1 and its role in Arabidopsis and other plants. The article references prior studies related to Karrikins and their signaling pathways, but a more extensive review of the literature and discussion of existing knowledge gaps would enhance the article. The study itself appears to be self-contained, presenting the identification and characterization of the SsSMAX1 gene in Sapium sebiferum. The results section provides data supporting the hypotheses related to the conservation of SsSMAX1 functions and its potential role in seed germination and seedling development in S. sebiferum. Overall, the text demonstrates a clear structure and provides relevant results supporting the hypotheses. However, later on I will provide some criticism and suggestions for the improvement of this article.

Experimental design

The objective of identifying and characterizing the SsSMAX1 gene suggests that there should be specific research questions or hypotheses driving the study. However, the article does not clearly outline a specific research question or hypothesis; instead, it describes the objectives and findings of the study. The study employed well-defined experimental procedures, including gene expression analysis and ectopic expression in Arabidopsis, which can be considered a strength. The cloning of the SsSMAX1 gene is described in detail, providing information on how the full sequence was obtained. The study also mentions the use of the floral dip method for the transformation of the recombinant expression vector in Arabidopsis, ensuring standardized plant growth conditions. However, the study lacks comprehensive step-by-step protocols or references to established methods for gene cloning and RNA extraction. Additionally, while the study mentions the use of statistical analyses with R Studio and provides the significance level used, it does not specify the specific statistical tests employed or provide details on how the data were analyzed and interpreted. Furthermore, the study does not mention the number of replicates used or whether the experiments were repeated independently, which is crucial for assessing the robustness and reliability of the results.

Validity of the findings

The article lacks an assessment of the impact and novelty of the research conducted in the SMAX text. It does not explicitly mention how the study contributes to the existing literature or addresses any gaps in knowledge. The absence of this assessment leaves the reader without a clear understanding of the significance of the research findings.
While the article encourages meaningful replication, it does not elaborate on its rationale or how it would benefit the scientific community. Without a clear statement of the rationale and benefits of the literature, evaluating the necessity and potential impact of replication studies becomes challenging.
Regarding data availability, the text states that all underlying data have been provided. It describes the data as robust, statistically sound, and controlled. However, the article does not provide specific details about the nature of the data, accesion numbers or the statistical analyses performed. This lack of information makes it difficult for readers to assess the reliability and validity of the findings presented in the study.
On a positive note, the conclusions in the article are well-stated and appear to be connected to the original research question. They are limited to supporting the results obtained in the study, which helps maintain the focus of the conclusions and their relevance to the research objectives. However, without a clear assessment of the impact and novelty of the research, the conclusions alone may not fully convey the significance of the findings.

Additional comments

Suggestions for improvements
The introduction has several areas that could be improved:
1. The introduction contains multiple abbreviations and acronyms, some without clear explanations (e.g., KAR/KL).
2. The information provided is presented in a somewhat disjointed manner, making it challenging to understand the flow of ideas, try to define each paragraph with a specific idea or a hidden title to make it clearer.
3. The references provided in the introduction are inconsistent in terms of citation format, and some are positioned after a full stop.
4. The introduction mainly summarizes previous research findings without offering any critical analysis or evaluation of the studies mentioned. Try to assess the strengths, limitations, and gaps in the existing literature to contextualize the current study.
5. It would be helpful to include a brief overview of the importance of seed germination and seedling development in plants and the relevance of KARs and SLs signaling pathways in these processes.

The M&M provides a clear description of the experimental procedures, allowing readers to understand the methods employed. However, there are a few areas that could be improved:
1. The MM lacks specific details regarding certain steps of the experimental procedures. For example, it would be helpful to provide more information on the primer design process, including the specific parameters and considerations used.
2. The references provided for the local blasting of Arabidopsis amino acid sequences and the construction of a local blast library are unclear. It would be beneficial to provide more specific information or clarify the sources of these references.
3. The MM does not mention the number of replicates, or the sample sizes used in the experiments.
4. It would be important to provide details about the control groups used in the experiments to ensure accurate interpretation of the results.
5. The authors used one-way analysis of variance (ANOVA) and post-hoc tests. However, it does not specify which tests were used for which data sets.
6. Ensure that all references are written in MM part. Recheck "Expression vector designing".

While the results provide valuable insights into the characteristics and potential functions of SsSMAX1, there are a few points of criticism:
1. The results primarily focus on qualitative observations and comparisons, lacking precise quantitative measurements.
2. The methods used for gene expression analysis, overexpression of SsSMAX1, and phenotypic measurements are not described in detail. Providing specific protocols and experimental details would improve reproducibility and allow other researchers to replicate the experiments.
3. The results mention the comparison between SsSMAX1, Atsmax1, and wild-type plants. However, it is unclear whether appropriate controls were included, such as transgenic lines with empty vectors or non-transformed plants.
4. The number of biological replicates or independent experiments conducted for each analysis is not mentioned.
5. The results do not mention any statistical tests or significant levels used to support the observed differences.

Discussion and conclusion:
1. The discussion does not provide a broader context for the findings. It does not explain why the study focused on SsSMAX1, how it contributes to the current understanding of Karrikin and SL signaling, or why Sapium sebiferum was chosen as the model plant.
2. Although the study refers to previous research, only a few references are provided; please include further references.
3. The discussion does not delve into the underlying molecular mechanisms through which SsSMAX1 affects plant development. Elaborating on the signaling pathways, transcriptional regulation, or protein interactions associated with SsSMAX1 would provide a deeper understanding of its function.
4. The conclusion states that SMAX1 has conserved functions in S. sebiferum, but it does not provide evidence or comparisons with other plant species to support this claim. Including comparative data from related species or highlighting similarities with previous studies would enhance the generalizability of the findings.
5. The conclusion focuses on the implications for forestry, horticulture, and the study of S. sebiferum specifically. However, it should refer to much broader implications or potential applications of understanding SMAX1 in other plant species or scientific fields.

Reviewer 2 ·

Basic reporting

- The manuscript does not present novel findings nor does it completely bring a significant advance to the field.

Experimental design

- The experimental design is disorganized. However, why two different mutants have been shown Atsmxl1 and Atsmax1?

Validity of the findings

- Not novel enough and does advance the field of plant biology significantly.

Reviewer 3 ·

Basic reporting

The authors successfully identified and characterized SsSMAX1, a gene in Sapium sebiferum, with significant commercial value as a bioenergy plant. Through a combination of sequence conservation analysis and RT-PCR technique, they were able to identify and clone the SsSMAX1 sequence. Gene expression analysis indicates that SsSMAX1 was highly expressed in the seed, hypocotyl, and root tips of the plant, suggesting its involvement in leaf shape regulation, hypocotyl development, and root architecture. To confirm the conservation of SsSMAX1's function, the researchers ectopically expressed the gene in Arabidopsis and compared its phenotypic effects with those of wild-type plants and AtSMAX1 functional mutants. The results demonstrated that overexpression of SsSMAX1 led to delayed seed germination, longer hypocotyls, longer root hairs, broader and longer leaves, and longer petioles, providing evidence for the functional conservation of SsSMAX1. The study utilizes bioinformatics analysis, gene expression profiling, and functional characterization to provide insights into the role of SsSMAX1 in Sapium sebiferum. The findings are supported by previous studies on AtSMAX1 and contribute to our understanding of the SMAX1-involved Karrikin signaling pathway in woody plants. The paper presents a straightforward rationale and a clean and tidy experimental design, which collectively enhance the field of Sapium sebiferum biology. The results offer unprecedented information for researchers working on Sapium sebiferum and other bioenergy plants. The methodology and results are clearly described, making this study a valuable reference for further research in this area.

Experimental design

Comments and suggestions regarding the paper’s strategy:
1. Have you investigated the in vivo expression of SsSMAX1 at the loci instead of relying on bioinformatic predicted ORFs? For example, numbers and expressions of transcripts at SsSMAX1 loci.
2. The overexpression strategy employed in the paper does offer supportive evidence to the conservation of SsSMAX1. However, it is worth considering that the overexpression of genes, especially when not in situ, can introduce confounding effects. To establish more direct and robust evidence, it would be beneficial to rescue an AtSMAX1 null mutant or perform an in situ replacement of AtSMAX1 with SsSMAX1. It would be interesting to understand why the authors chose to begin with an overexpression strategy.
3. The gene cloning of SsSMAX1 represents a crucial aspect of the study. To enhance the clarity and impact of the paper, providing additional figures or detailed descriptions regarding the gene cloning process would be valuable. This would further contribute to the overall quality and presentation of the research findings.

Validity of the findings

The detailed comments and suggestions:
• Line 29, delete “application”.
• Line 32, use the complete spelling of “SLs” when it appears for the first time in the paper.
• Line 34, briefly explain “catalytic triad” to readers.
• Line 42, replace “the phenotype differences between…” with “the differences between…phenotypes”.
• Line 44, explain what function the “MAX2 functional mutant” serves.
• Line 44, explain “combination of KAI2 and D14 effects” in more detail.
• When citing the findings/results of a published paper, describe the observations and experiments conducted in the paper that led to the conclusion. This provides necessary background information and avoids misleading the readers. This applies to the entire first paragraph of the introduction.
• Line 72, italicize “SSMAX1”.
• Figure 1, improve the resolution.
• Figure 1B, describe how 3D structure of SsSMAX1 was obtained. Also add this information to the method part of the paper.
• Figure 1D, check the label to see if it’s “PtSMAX1” or “PaSMAX1”.
• Line 140, clarify which protein alignment matrix was used to determine the 65.1% similarity.
• Line 142, explain how the phylogenetic tree was constructed.
• Line 144, discuss any biological significance behind SsSMAX1 being closer to PtSMAX1 in K signaling compared to others.
• Line 161, replace “performed” with “evaluated”.
• Line 156, explain the genotype of Atsmax 1 functional mutant.
• Line 182, delete “these” or “both”.
• Line 207, replace “more” with “longer”.

Annotated reviews are not available for download in order to protect the identity of reviewers who chose to remain anonymous.

---

## Round 0.2 · Minor Revisions

Dear Authors

The manuscript still needs a minor revision to be reconsidered for publication. The authors are invited to revise the paper considering all the suggestions made by the reviewers. Significant concerns about the manuscript's grammar, usage, and overall readability exist. Therefore, revise the text to fix grammatical errors and improve the text's overall readability. We suggest you have a fluent, preferably native, English-language speaker thoroughly copyedit your manuscript for language usage, spelling, and grammar. If you do not know anyone who can do this, PeerJ can provide language editing services. Please note that requested changes are required for publication.

With Thanks

**Language Note:** The Academic Editor has identified that the English language must be improved. PeerJ can provide language editing services - please contact us at [email protected] for pricing (be sure to provide your manuscript number and title). Alternatively, you should make your own arrangements to improve the language quality and provide details in your response letter. – PeerJ Staff

·

Basic reporting

The authors have diligently addressed the raised concerns, encompassing a comprehensive range of modifications. This includes thoughtful incorporation of numerous suggestions and recommendations put forth during the review process.

Experimental design

The authors have taken substantial steps to address the critiques concerning the experimental design that were previously raised. Their efforts are reflected in the significant modifications made, which have incorporated numerous suggestions and recommendations provided during the initial review.

Validity of the findings

The authors have made noteworthy revisions to enhance the validity of their findings, effectively incorporating the feedback and suggestions previously offered. These improvements demonstrate a commendable dedication to strengthening the credibility and robustness of their research outcomes.

Additional comments

The article necessitates both technical and linguistic revisions, while it is evident that the formatting does not align with the guidelines outlined by PeerJ.

Reviewer 3 ·

Basic reporting

The authors successfully identified and characterized SsSMAX1, a gene in Sapium sebiferum, with significant commercial value as a bioenergy plant. Through a combination of sequence conservation analysis and RT-PCR technique, they were able to identify and clone the SsSMAX1 sequence. Gene expression analysis indicates that SsSMAX1 was highly expressed in the seed, hypocotyl, and root tips of the plant, suggesting its involvement in leaf shape regulation, hypocotyl development, and root architecture. To confirm the conservation of SsSMAX1's function, the researchers ectopically expressed the gene in Arabidopsis and compared its phenotypic effects with those of wild-type plants and AtSMAX1 functional mutants. The results demonstrated that overexpression of SsSMAX1 led to delayed seed germination, longer hypocotyls, longer root hairs, broader and longer leaves, and longer petioles, providing evidence for the functional conservation of SsSMAX1. The study utilizes bioinformatics analysis, gene expression profiling, and functional characterization to provide insights into the role of SsSMAX1 in Sapium sebiferum. The findings are supported by previous studies on AtSMAX1 and contribute to our understanding of the SMAX1-involved Karrikin signaling pathway in woody plants. The paper presents a straightforward rationale and a clean and tidy experimental design, which collectively enhance the field of Sapium sebiferum biology. The results offer unprecedented information for researchers working on Sapium sebiferum and other bioenergy plants. The methodology and results are clearly described, making this study a valuable reference for further research in this area.

Experimental design

The strategy employed in the paper does offer supportive evidence to test the conservation of SsSMAX1.

Validity of the findings

The reliability of the findings has been thoroughly validated.

Additional comments

While the authors have not addressed the question regarding the specific types of in vivo transcripts that could be experimentally identified, as raised in the inquiry "Have you investigated the in vivo expression of SsSMAX1 at the loci instead of relying on bioinformatic predicted ORFs? For instance, considering the quantity and expression levels of transcripts at SsSMAX1 loci," it's worth noting that delving into this aspect would likely constitute an independent project. As a possible avenue for future research, the authors are encouraged to consider planning and executing an investigation in this direction. In conclusion, the authors have successfully incorporated the reviewer's feedback and carried out appropriate revisions in the manuscript.

---

## Round 0.3 · Minor Revisions

The authors have made noteworthy revisions to enhance the manuscript, but still need linguistic revisions. Again, we suggest you have a fluent English-language speaker thoroughly copyedit your manuscript for language usage, spelling, and grammar. If you do not know anyone who can do this, PeerJ can provide language editing services. Please note that requested changes are required for publication.

Moreover, the formatting still does not align with the guidelines outlined by PeerJ. For example, why all the references are in italics lines? Furthermore, please check that all the genes' names are in italic lines.

**Language Note:** The Academic Editor has identified that the English language must be improved. PeerJ can provide language editing services - please contact us at [email protected] for pricing (be sure to provide your manuscript number and title). Alternatively, you should make your own arrangements to improve the language quality and provide details in your response letter. – PeerJ Staff

---

## Round 0.4 · accepted · Accept

Dear Authors

I am pleased to inform you that after the last round of revision, the manuscript has been improved a lot, and it can be accepted for publication.

Congratulations on the acceptance of your manuscript, and thank you for your
interest in submitting your work to PeerJ.